# Immunohistochemical Profiling of PD-1, PD-L1, CD8, MSI, and p53 and Prognostic Implications in Advanced Serous Ovarian Carcinoma: A Retrospective Study

**DOI:** 10.3390/jpm13071045

**Published:** 2023-06-26

**Authors:** Ana Paula Dergham, Caroline Busatta Vaz de Paula, Seigo Nagashima, Márcia Olandoski, Lucia de Noronha, Vanessa Santos Sotomaior

**Affiliations:** 1Graduate Program in Health Sciences, School of Medicine and Life Sciences, Pontifícia Universidade Católica do Paraná (PUCPR), Curitiba 80215-901, Brazil; carolbvaz@gmail.com (C.B.V.d.P.); seigocap@gmail.com (S.N.); marcia.olandoski@pucpr.br (M.O.); lucia.noronha@pucpr.br (L.d.N.); 2Neo Oncologia Núcleo de Estudos Oncológicos, Curitiba 80440-210, Brazil

**Keywords:** biomarkers, immunotherapy markers, advanced serous ovarian carcinoma, immunohistochemistry, p53, MSI

## Abstract

Advanced high-grade serous ovarian carcinoma is a serious malignant neoplasm with a late diagnosis and high mortality rate. Even when treated with standard therapy, such as surgery followed by carboplatin and paclitaxel chemotherapy, the prognosis remains unfavorable. Immunotherapy is a treatment alternative that requires further study. Therefore, we aimed to evaluate the expression of PD-1, PD-L1, CD8, MSI (MLH1, MSH2, MSH6, and PMS2), and p53 in the paraffin samples of high-grade serous ovarian carcinoma. A retrospective study of 28 southern Brazilian patients with advanced serous ovarian carcinoma (EC III or IV) was conducted between 2009 and 2020. The expression of these proteins was evaluated using immunohistochemistry, and the results were correlated with the patients’ clinicopathological data. At diagnosis, the mean age was 61 years, and the most common clinical stage (60%) was EC III. Among the cases, 84.6% exhibited p53 overexpression, 14.8% had MSI, 92.0% were sensitive to platinum, and more than 50.0% relapsed after treatment. Patients with MSI had a lower CD8/PD-1 ratio and more relapses (*p* = 0.03). In conclusion, analysis of immunotherapeutic markers in paraffin-embedded advanced serous ovarian carcinoma samples is feasible and may assist in prognosis.

## 1. Introduction

Advanced high-grade serous ovarian carcinoma is women’s most common and deadliest subtype of ovarian cancer [1]. Among Brazilian women, ovarian cancer is the eighth most common type of cancer, with 7310 new cases estimated in 2022, and, as in the rest of the world, presents a worrying proximity between incidence and death rates per 100,000 inhabitants [2].

Despite advances in therapy, recurrence is common. Loss of sensitivity to platinum, the main drug used for treatment, correlates with poor prognosis. In the last 10 years, there has been an increase in survival with therapies including anti-VEGF (vascular endothelial growth factor) drugs (bevacizumab) [3] and PARP (poly (ADP-ribose) polymerase 1) enzyme inhibitors (olaparib, rucaparib, veliparib, and niraparib) [4,5,6]. However, maximal cytoreduction (debulking) remains the most significant prognostic factor for increased survival in all stages of ovarian cancer [7]. Moreover, the high degree of molecular heterogeneity and expression of various neoantigens [8,9] hinder the progress of ovarian cancer therapy.

Therefore, immunotherapy has emerged as an alternative treatment. This drug class has been studied and validated for other types of cancer, such as melanoma and lung cancer [10,11,12,13,14,15].

The results of the first studies on ovarian cancer with immunotherapies alone and/or in combination with conventional drugs are beginning to emerge [9,16,17,18]; however, identifying patients who can benefit from this treatment needs to be better established.

To the best of our knowledge, there has been no validation of biomarkers as prognostic, predictive of response, or an indication for immunotherapy for high-grade serous ovarian cancer. Biomarkers such as PD-1 (programmed cell death 1 or PDCD1), PD-L1 (programmed cell death 1 ligand 1 or CD274 molecule), and CD8, in addition to microsatellite instability (MSI), have been recognized as valuable markers for indicating immunotherapy in other types of cancer [19,20].

MMR (mismatch repair) corresponds to a family of genes related to functional loss and DNA repair: *MLH1* (mutL homolog 1), *MSH2* (mutS homolog 2), *MSH6* (mutS homolog 6), *PMS2* (postmeiotic segregation increased 2) and *MLH3* (mutL homolog 3) [21]. Mutations, including inherited or somatic deletions in both alleles of these genes, lead to impaired genomic repair mechanisms during DNA replication in the S phase of the cell cycle. This confers a high propensity for the accumulation of mutations in cells. This genetic failure can be detected by investigating MSI [22]. Microsatellites are small regions of the genome in which nucleotides are sequentially repeated. In these regions, replication errors are more easily recognized because the repetitions favor the occurrence of point mutations as well as small deletions and insertions near the repeat sequences. MSI analysis can be performed either by a polymerase chain reaction (PCR) or indirectly by immunohistochemistry.

In the tumor microenvironment, the binding between PD-L1 on tumor cells and PD-1 on T lymphocytes allows the tumor to escape the immune response by inactivating cytotoxic CD8+ T cells. Thus, blocking this binding using specific antibodies to PD-L1 or PD-1 reverses this condition, and the T cell lyses the neoplastic cell because it now identifies it as foreign [23].

The premise for the use of immunotherapy in epithelial ovarian cancer is the high expression of PD-L1 in tumor cells, greater than or equal to 1% [17]. MSI is found in approximately 10% of ovarian cancer cases, particularly the endometrioid and mucinous subtypes [20]. The prognosis of sporadic tumors exhibiting MSI is favorable. These tumors respond better to immunotherapy [24,25,26], and the presence of MSI is considered a marker and predictor of response to immunotherapy [19,20].

The p53 protein is encoded by the tumor suppressor gene *TP53* (tumor protein p53). It regulates the expression of various genes in response to different types of cellular stress, inducing cell cycle arrest, apoptosis, cellular senescence, DNA repair, and changes in metabolism. The expression of p53 is altered in high-grade carcinomas, appearing as either completely negative or diffusely positive and overexpressed. In the latter case, the protein is mutated [27]. Mutations in *TP53* define the characteristics of high-grade serous carcinomas and occur in 96% of these cases [28,29].

To optimize this therapy, especially for platinum-resistant and refractory tumors, which lead to death in approximately 1 year, and platinum-sensitive tumors, which relapse in 95% of cases, it is necessary to understand the markers of immunotherapy and predictors of response better, especially in the most common subtype, serous epithelial ovarian tumors.

The objective of this study was to analyze, by immunohistochemistry, the expression of PD-1, PD-L1, CD8, MLH1, MSH2, MSH6, PMS2, and p53 in paraffin-embedded tumor samples from patients with advanced serous ovarian carcinoma and to correlate these expression profiles with the clinicopathological data of the patients.

## 2. Materials and Methods

### 2.1. Patients and Samples

The samples included in this retrospective study were obtained from the pathology services of two hospitals in Curitiba, Brazil, with the approval of their respective research ethics committees. These tumor samples were obtained during surgical procedures from 2009 to 2020 to remove primary ovarian tumors from patients who had not undergone neoadjuvant chemotherapy.

Thus, a convenience sample of 28 patients with advanced serous ovarian carcinoma at clinical stage III or IV (EC III or IV) was retrospectively analyzed. These patients presented with paraffin blocks suitable for the proposed study and available data in medical records. Twelve patients were excluded as they did not meet these criteria.

Data from the following clinical variables of the patients and pathological variables of the samples were recorded and analyzed: diagnosis, age at diagnosis, body mass index (BMI), Eastern Cooperative Oncology Group (ECOG) performance status [30], TNM staging, which includes the extent of the tumor (T), the extent of spread to the lymph nodes (N), presence of metastasis (M) [31], history of previous cancer, presence of mutations in the BRCA1 and BRCA2 genes, associated comorbidities, recurrence, site of recurrence, recurrence-free survival, and overall survival. Survival data were updated in November 2021.

### 2.2. Histological Analysis

The anatomopathological patterns of the samples were reviewed by an experienced pathologist (L. de N.) on slides stained with hematoxylin and eosin (Harris Hematoxylin: NewProv, Cod. PA203, Paraná, Brazil; Eosin: BIOTEC Reagentes Analíticos, Cod. 4371, Paraná, Brazil).

### 2.3. Immunohistochemical Tests

Immunohistochemical analysis was used to determine the expression of PD-1, PD-L1, CD8, MLH1, MSH2, MSH6, PMS2, and p53 in the paraffin-embedded tumor samples.

### 2.4. PD-1, PD-L1, CD8, and p53

The immunohistochemical assay used a protocol of incubation of specific primary antibodies for PD-1 (J116/14-9989-82; Thermo Fisher (Waltham, MA, USA); dilution 1:100), PD-L1 (PA5-28115; Thermo Fisher; dilution 1:200), CD8 (SP16; Thermo Fisher; dilution 1:100), and p53 (D07/BSB5844; BioSB; dilution 1:200) in a humid chamber at a temperature between 2 and 8 °C, overnight. The secondary polymer, mouse-, and rabbit-specific HRP/DAB IHC Detection Kit (Micro-polymer, ab236466 Abcam, Cambridge, UK) was applied to the test material for 25 min at room temperature. The technique was revealed by adding a 2,3-diaminobenzidine and hydrogen peroxide substrate complex for sufficient time for it to develop a brown color, followed by counterstaining with Harris hematoxylin.

The positive controls for these reactions were determined by the immunopositivity of tissue samples that were reactive to the antibodies in the test. These samples were placed on slides along with the studied samples. Lymph node samples for CD8 and breast carcinoma samples for p53 were analyzed together with the reactions of the tested samples. The negative control consisted of omitting the primary antibody from the reaction.

### 2.5. MLH1, MSH2, MSH6 and PMS2

The immunohistochemical assay consisted of a protocol of incubation of primary antibodies specific for MLH1 (ES05; Dako (Agilent Technologies, Santa Clara, CA, USA); dilution 1:200), MSH2 (FE11; Dako; ready to use), MSH6 (EP49; Dako; ready to use), and PMS2 (EP51; Dako; ready to use) in a humid chamber at a temperature between 2 and 8 °C, overnight. A secondary polymer (Dako EnVisionTM FLEX/HRP, Santa Clara, CA, USA) was applied to the test material for 30 min at room temperature. The technique was revealed by the addition of the complex 2, 3, diaminobenzidine and hydrogen peroxide substrate for sufficient time to develop a brown color, followed by counterstaining with Harris hematoxylin.

The positive controls for these reactions were determined based on the reactivity of human colon tissue samples that were reactive to the tested antibodies. These samples were then allocated to slides, together with the studied samples. The negative control consisted of omitting the primary antibody from the reaction.

### 2.6. Tissue Immunoexpression Analysis

#### 2.6.1. PD-L1

Immunolabeled slides with specific antibodies against PD-L1 were scanned using an Axio Scan.Z1 slide scanner (Zeiss, Jena, Germany). Thirty high-magnification field images, 40×, were generated using the ZEN Blue Edition software (Zeiss).

The analyses were performed blindly using images obtained from random sample regions without interference from an observer. In each image, the areas of immunopositivity were measured using Image Pro-Plus software version 4.5 (Media Cybernetics, Rockville, MD, USA) using a semi-automated color segmentation method, in which the PD-L1 immunopositive area was delimited and quantified.

Next, the value of the immunopositive area, expressed in square micrometers (μm^2^), was divided by the total tumor area and transformed into a percentage value. Finally, the arithmetic mean values of the images were calculated for each patient.

#### 2.6.2. CD8 and PD-1

Immunostained slides with specific antibodies against PD-1 and CD8 were scanned using an Axio Scan.Z1 slide scanner (Zeiss, Jena, Germany). The digitized files were visualized using ZEN Blue Edition software (Zeiss, Jena, Germany), and a rectangle with an area of 1 mm^2^ was selected and positioned over the hotspot regions (areas with a high density of lymphocytes); after that, the lymphocytes were counted for each immunostained antibody.

#### 2.6.3. MLH1, MSH2, MSH6, and PMS2

Slides immunostained with anti-MLH-1, MSH-2, MSH-6, and PMS-2 antibodies were analyzed using a BX40^®^ microscope (Olympus, Tokyo, Japan) at 40× magnification and paired as follows: MSH2 + MSH6 and MLH1 + PMS2. Each sample that comprised a case was analyzed. At least one positive area was classified as positive, and the absence of any positive area was classified as negative for the antibody in question. Patients were defined as MSI carriers when at least one of the four MMR proteins (MLH1, MSH2, MSH6, or PMS2) tested negative.

#### 2.6.4. p53

Slides immunostained for p53 were analyzed under a BX40 microscope (Olympus, Tokyo, Japan) at 40× magnification (Figure 1). The expression of p53 was classified as overexpression (mutated) or wild type (non-mutated) in the samples studied.

### 2.7. Statistical Analysis

Quantitative variables are described as means and standard deviations. Categorical variables are expressed as frequencies and percentages. The Student’s *t*-test for independent samples or the non-parametric Mann–Whitney U test was used to compare two groups of dichotomous categorical quantitative variables. The association between two categorical variables was analyzed using Fisher’s exact test. The Cox Regression model and log-rank test were used to examine factors associated with recurrence-free and overall survival. The normality of continuous quantitative variables was evaluated using the Shapiro–Wilk test. Statistical significance was set at a *p*-value of less than 0.05. Data were analyzed using Stata/SE software version 14.1 (StataCorp LP, College Station, TX, USA).

## 3. Results

All 28 patients were Caucasian and had an average age of 61.3 ± 14 years at diagnosis. The youngest and oldest patients were 34 years old and 90 years old, respectively. Patients were generally observed to be overweight, with an average BMI of 25.9 ± 5.8 (Table 1).

Most of the patients (*n* = 21) had high-grade serous epithelial carcinoma. Eight patients had stage IV disease at diagnosis, and 19 had stage III disease. Seven patients reported a history of cancer, three of whom had breast cancer. Most patients did not have mutations in *BRCA1* (11/18) or *BRCA2* (12/18), but three had germline and somatic mutations in *BRCA1* and *BRCA2*.

These three patients were sensitive to platinum and had wild type p53 (different from most cases in the study). One patient was assigned to the MSI-unstable group. All three patients had lower PD-1 expression than the average in the studied group. Similarly, for PD-L1, two patients had below-average expression and only one had a CD8 count within the median value. These three patients’ CD8/PD1 ratios were above the study’s average.

Most patients (17) underwent cytoreduction rather than surgical biopsy. The most used chemotherapy regimen (21 patients) was a combination of carboplatin and paclitaxel, and 23 patients were sensitive to platinum. The median overall survival was 50.8 months, and the median progression-free survival was 17.2 months. Fifteen patients relapsed and six died during the study period (Table 2).

The results of tissue immunoexpression analysis are presented in Figure 1 and Figure 2 and Table 3.
Figure 1Tissue immunoexpression of PD-1, PD-L1, p53, and CD8 in samples of high-grade serous carcinoma (PD1 and CD8). The evaluation of tissue immunoexpression of CD-8 and PD-1 was performed from the nuclear count (hotspots) of labeled lymphocytes, as evidenced by their brown staining (p53). Tissue immunoexpression of p53 was evaluated through the immunohistochemical expression of an anti-p53 antibody. Positive staining was determined from the nuclear count of immunolabeled cells, as evidenced by the brown staining of the antigen-antibody reaction (PD-L1). Evaluation of the tissue expression of PD-L1 was performed through morphometric analysis, as evidenced by the areas observed by brown staining. Photomicrograph at 40× magnification.
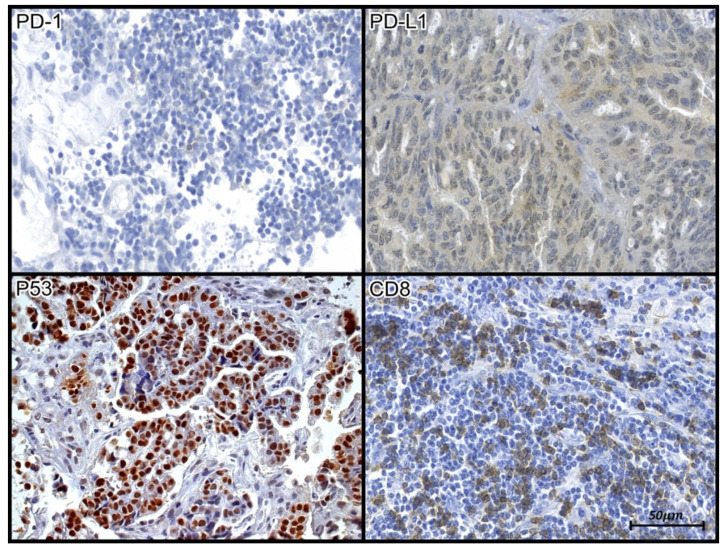



The expression profiles of PD-1 and PD-L1 did not correlate with each other (r = 0.10, *p* = 0.642), nor were their profiles, as well as those of p53, associated with the grade and clinical stage of serous ovarian carcinoma (*p* > 0.05).

The MSI status was compared with the PD-1, PD-L1, and CD8/PD-1 ratio expression. There was a trend towards a higher CD8/PD-1 ratio in patients with stable MSI (*p* = 0.049) (Table 4), although only four patients showed instability. MSI status was not associated with serous ovarian carcinoma grade, clinical stage, or platinum sensitivity (*p* > 0.05); however, it did seem to affect recurrence-free survival (Table 5; Figure 3). Furthermore, there was no significant association of the studied variables with death in this sample.
jpm-13-01045-t004_Table 4Table 4Association of MSI status with PD-1 and PD-L1 expression and CD8/PD-1 ratio.MarkerStatus of MSI*n* ^1^Median (Min–Max)IQR ^2^*p* *PD-1stable2316.0 (1.0–62.0)29.00.272unstable46.0 (3.0–17.0)9.0CD8/PD-1stable2312.5 (0.1–77.0)37.1**0.049**unstable41.8 (0.7–7.6)4.5PD-L1stable229.8 (0.5–72.4)11.50.471unstable45.0 (0.8–51.3)27.6*n* = number of cases. * Non-parametric Mann–Whitney test, *p* < 0.05 indicates statistical significance. ^1^ Number of cases differs in the variables due to failures in the patient records. ^2^ IQR: interquartile range (difference between quartiles 1 and 3).
jpm-13-01045-t005_Table 5Table 5Association of recurrence with p53 expression, MSI status, clinical stage, and grade of serous carcinoma.Variable*n* ^1^Relapse*p* **n* (%)p53overexpressed2212 (54.5)0.496wild type42 (50.0)MSIstable2310 (43.5)**0.031**unstable44 (100.0)Clinical stageIII199 (47.4)0.948IV86 (75.0)Grade of serous carcinomahigh2113 (61.9)0.862low72 (28.6)*n* = number of cases. ^1^ Number of cases differs in the variables due to failures in the patient records. * Log-rank test, *p* < 0.05 indicates statistical significance.


## 4. Discussion

The prognosis of advanced ovarian cancer is poor, with approximately only 20% of patients surviving 5 years after diagnosis [32]. The standard therapy of complete cytoreduction combined with chemotherapy has not been sufficient for more advanced stages, as most patients develop platinum resistance within 16–18 months of treatment [33]. The data from this study showed a progression-free survival of 17.2 months.

A previous study found that age of more than 60 years old was an independent factor associated with worse survival [32], and most patients (68%) in this study were diagnosed at more than 60 years of age. Additionally, the clinical stage was an independent prognostic factor for shorter survival. Ovarian cancer is usually diagnosed at stages III and IV, with approximately 30% of patients presenting with metastasis at diagnosis (stage IV), indicating inoperability and poor prognosis.

*TP53* expression is an important prognostic indicator of malignant neoplasms. Epithelial ovarian cancer is mutated in 40–80% of cases. In a previous study of 105 patients with ovarian cancer, mutations were found in approximately 57% of cases [34]. Alterations in p53 levels are associated with poorly differentiated disease, platinum resistance, early relapse, and decreased overall and disease-free survival [34]. Immunohistochemical analysis revealed that 85% of the cases overexpressed p53, indicating a mutation. Moreover, a few studies quantitatively correlated p53 expression with survival in ovarian cancer [35].

The frequency of MSI in ovarian cancer varies from 2 to 20% [36,37]. Most MSI cases are clear cell and endometrioid carcinoma types [38]. Notwithstanding that, in this study, 14.8% of patients had MSI; they were high-grade serous adenocarcinoma cases.

Yamashita et al. [39], in a review of 136 cases, found that 4.4% of MSI cases included endometrioid, mucinous, and clear cell carcinoma subtypes. High-grade serous carcinoma presented only two MSI cases out of the 67 studied. In line with Yamashita, there was no significant association between MSI and clinical stage or expression of the immune markers PD-1, PD-L1, or CD8 in this analysis.

PD-L1 expression in tumor cells is the main strategy for immune evasion in cancer. Higher PD-L1 expression is associated with lower overall survival [16]. This is due to a reduction in T-lymphocyte infiltration into the tumor, suggesting that PD-L1 expression promotes an immunosuppressive microenvironment by inhibiting lymphocyte infiltration. Hamanishi et al. [16] were the first to describe PD-L1 expression in ovarian cancer and found expression in 88% of the tumor cells. The authors demonstrated an inverse relationship between PD-L1 expression and the number of CD8+ lymphocytes. Women with high PD-L1 expression have worse overall and progression-free survival [40].

Ovarian cancer is a complex disease with unique characteristics and immune microenvironments. Studying PD-1 and PD-L1 specifically in the context of advanced serous ovarian carcinoma allows exploration of their expression patterns and potential clinical implications in this subset of patients.

CD8-positive T lymphocytes play a crucial role in the immune response against tumors. CD8-positive T cells are primarily cytotoxic T cells that directly target and eliminate cancer cells. Evaluating the presence and abundance of CD8-positive TILs can provide insights into the anti-tumor immune response and potential prognostic implications in ovarian cancer.

In this study, all patients had low percentages of CD8/PD-1 ratios (median of 11%), similar to the immunosuppressive microenvironment described in the literature. Furthermore, patients with stable MSI had higher CD8/PD-1 ratios.

The PD-1/PD-L1 pathway is an immune escape phenomenon in tumors that has not yet been validated as a biomarker for ovarian cancer. There are no predictive or prognostic biomarkers to determine which patients would benefit from ovarian cancer immunotherapy.

The isolated expression of PD-L1 cannot be used as a potential indicator of the benefits of PD-1/PD-L1 inhibitors in the absence of TILs (tumor-infiltrating lymphocytes) [41]. Zhang et al. [42] analyzed 174 patients and showed that the presence of TILs was associated with increased overall survival in ovarian cancer compared with a cohort without TILs. These data are consistent with other studies summarized in a meta-analysis of 1815 patients [43]. This positive prognostic effect can be attributed to the subgroup of intratumoral CD8+ T cells.

A limitation of this study was the small sample size (some blocks were lost because of poor material quality). Nevertheless, the statistically significant demonstration related to the association between recurrence and MSI status should be considered in the context of the sample size in this study (*n* = 4). Moreover, it should also be noted that the samples were exclusively serous carcinomas, without other histologies, as reported in previous studies. However, we cannot exclude the possibility that a larger sample size may yield different results.

Furthermore, making inferences based solely on the results presented here, that the presence of MSI impacts recurrence-free survival, may seem ambitious. According to robust data from the literature, malignant tumors with unstable MSI respond better to immunotherapy [44].

Given tumor heterogeneity, the selection of patients is essential for effective immunotherapy in high-grade epithelial cancer. This selection can also optimize cancer treatment costs by providing specific therapies to selected patients.

## 5. Conclusions

The clinical and histopathological characteristics of the studied samples corroborated data from the literature. Samples with MSI stability showed a higher proportion of CD8/PD-1 cases, and patients with this condition had less recurrence. In contrast, there was an association between recurrence and MSI instability (*p* = 0.03).

The present study aimed to evaluate comprehensively the expression of multiple immune markers, including PD-1, PD-L1, and CD8, in advanced serous ovarian carcinoma. By including CD8 as a specific focus, we intended to investigate the role of cytotoxic T cell responses and their potential correlation with the expression of immune checkpoint markers. This approach enables a more comprehensive understanding of the immune landscape and its possible implications in this specific ovarian cancer subtype.

Developing immune checkpoint inhibitors targeting PD-1 and PD-L1 has revolutionized cancer therapy in recent years. While these agents have shown promising results in various malignancies, the response rates and clinical outcomes in ovarian cancer may differ. Therefore, understanding the expression levels and clinical relevance of PD-1 and PD-L1 in ovarian cancer can aid in identifying potential responders to immunotherapy and optimizing treatment strategies.

## Figures and Tables

**Figure 2 jpm-13-01045-f002:**
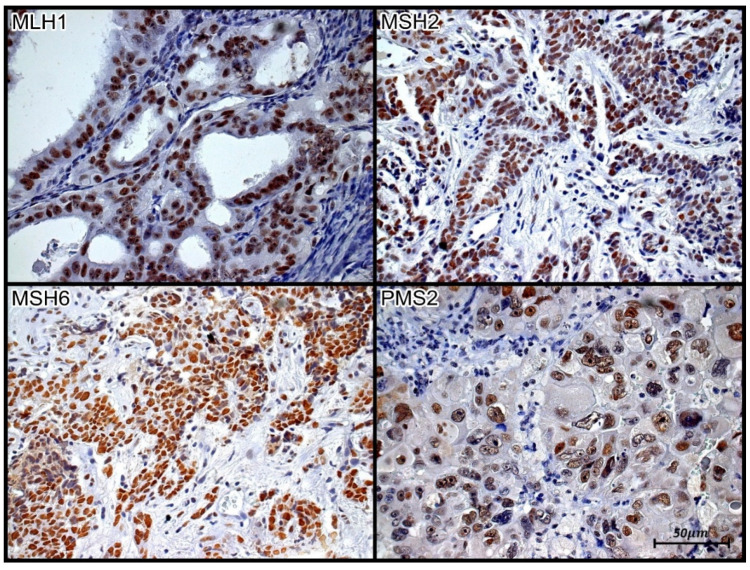
Tissue immunoexpression of MLH1, MSH2, MSH6, and PMS2 in high-grade serous ovarian carcinoma. Brown nuclear staining represents tissue immunopositivity for the immunohistochemical reaction with the monoclonal antibodies anti-MLH-1, anti-MSH2, anti-MSH6, and anti-PMS2, demonstrating the presence of the respective proteins in the nuclei of the papillary serous ovarian carcinoma cells. Immunolabeling of at least one cell was considered positive in the evaluated case. Photomicrograph at 40× magnification.

**Figure 3 jpm-13-01045-f003:**
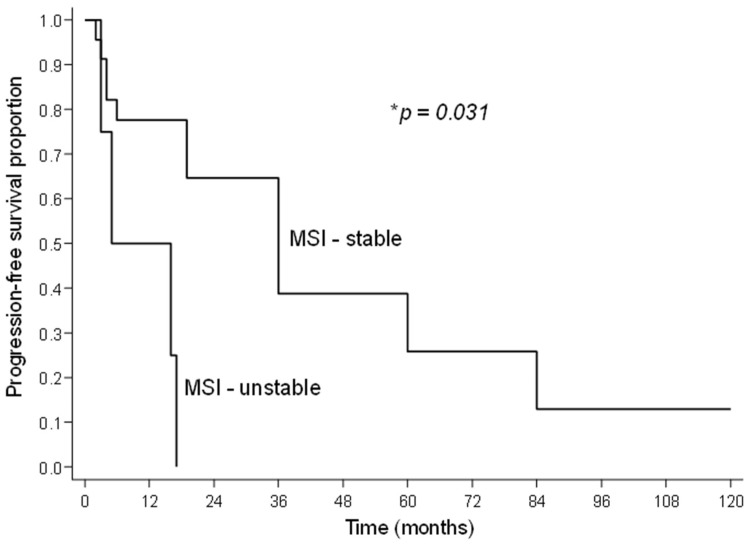
Curve showing the differences in recurrence between patients with different MSI statuses. * Log-rank test, *p* < 0.05 indicates statistical significance.

**Table 1 jpm-13-01045-t001:** Quantitative clinical–pathological characterization of the patients.

Variable	*n* ^1^	Average ± SD	Median (Min–Max)	IQR
Age at diagnosis (years)	28	61.3 ± 14.0	60.0 (34.0–90.0)	18.0
BMI	26	25.9 ± 5.8	25.2 (18.6–41.3)	6.3

*n*, number of cases; SD, standard deviation; Min, minimum value; Max, maximum value; IQR, interquartile range (difference between quartiles 1 and 3); and BMI, body mass index. ^1^ The number of cases differs between variables due to missing patient medical records data.

**Table 2 jpm-13-01045-t002:** Clinicopathological characteristics of the cases.

Variable		*n* ^1^	%
Diagnosis ^2^	high grade	21	75.0
	low grade	7	25.0
Size of the tumor	T3	24	96.0
Compromise of lymph nodes	no	11	42.3
	yes	15	57.7
Metastasis	no	18	69.2
	yes	8	30.8
Clinical stage	III	19	70.4
	IV	8	29.6
History of cancer	no	13	65.0
	yes	7	35.0
*BRCA1* mutation	no	11	61.1
	yes	7	38.9
*BRCA2* mutation	no	12	66.7
	yes	6	33.3
*BRCA* mutation type	germline	7	70.0
	tumor	3	30.0
Comorbidity	no	10	38.5
	yes	16	61.5
Platinum sensitivity	no	2	8.0
	yes	23	92.0
Relapse	no	13	46.4
	yes	15	53.6
Outcome	dead	6	21.4
	alive	18	64.3
	segment loss	4	14.3

*n* = number of cases; T3 = tumor affecting one or both ovaries and/or tubes with confirmed microscopic disease outside the pelvis and/or retroperitoneal metastasis (pelvic and/or para-aortic lymph nodes). ^1^ Number of cases differs in the variables due to failures in the patients’ medical records. ^2^ Corresponds to the surgical pathological diagnosis.

**Table 3 jpm-13-01045-t003:** Tissue immunoexpression of biomarkers in advanced serous ovarian carcinoma samples.

Marker	*n* ^1^	Result
PD-1	28	18.8 ± 17.2
CD8	28	346.6 ± 469.2
PD-L1	26	15.2 ± 18.1
MLH1	5	Negative
	23	Positive
MSH2	2	Negative
	26	Positive
MSH6	1	Negative
	27	Positive
PMS2	3	Negative
	25	Positive
p53	6	wild type
	22	overexpressed
status MSI	23	stable
	4	unstable

*n* = number of cases. ^1^ Number of cases differs in the variables due to failures in the patient records.

## Data Availability

The data presented in this study are available on request from the corresponding author. The data is not publicly available due to privacy restrictions.

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
