# Peer review of "Immunohistochemical Profiling of PD-1, PD-L1, CD8, MSI, and p53 and Prognostic Implications in Advanced Serous Ovarian Carcinoma: A Retrospective Study"

_jpm, 2023, doi:10.3390/jpm13071045_

Round 1

Reviewer 1 Report

The manuscript presents the results on the role of different markers in advanced serous ovarian carcinoma. These are interesting results, however, there are several major issues:

1. The title should be rephrased since CD8, MSI and p53 are not immune chekpoints

2. There are several immune checkpoints in T cells and tumor cells, why authors decided to study PD-1 and PD-L1, there are already a lot of studies assessing these markers. 

3. Why authors studied CD8? And not CD4? Or other T cell markers? Are there no piblished studies assessing CD8 in advanced serous ovarian cancer?

4. Did authors measured the level of immune checkpoints in peripheral blood lymphocytes? 

5. Authors should clearly state which results in the study are novel 

Author Response

Dear Reviewer 1,

We sincerely appreciate your time and effort in reviewing our scientific article. Your thoughtful input has been precious in improving our research and presenting it more clearly and coherently.

While we acknowledge the modest nature of our article, your critical feedback has played a significant role in enhancing its overall quality. We are grateful for the constructive suggestions that have strengthened our research and refined our findings.

Below are our point-by-point responses addressing the issues you raised.

  1. The title should be rephrased since CD8, MSI and p53 are not immune checkpoints.

After carefully considering your suggestion, we decided to change the title to "Immunohistochemical Profiling of PD-1, PD-L1, CD8, MSI, and p53 and Prognostic Implications in Advanced Serous Ovarian Carcinoma: A Retrospective Study." We aimed to create a title that better captures the various aspects of our study and its potential impact on prognosis.

  1. There are several immune checkpoints in T cells and tumor cells, why authors decided to study PD-1 and PD-L1, there are already many studies assessing these markers.

While it is true that PD-1 and PD-L1 have been extensively investigated in various cancer types, including ovarian cancer, their significance and potential as therapeutic targets remain a subject of ongoing research and clinical interest. Despite the existing body of literature, several aspects justify our study:

Contextual Relevance: Ovarian cancer is a complex disease with unique characteristics and immune microenvironments. Studying PD-1 and PD-L1 specifically in the context of advanced serous ovarian carcinoma allows us to explore their expression patterns and potential clinical implications in this subset of patients.

Treatment Implications: Developing immune checkpoint inhibitors targeting PD-1 and PD-L1 has revolutionized cancer therapy in recent years. While these agents have shown promising results in various malignancies, the response rates and clinical outcomes in ovarian cancer may differ. Therefore, understanding the expression levels and clinical relevance of PD-1 and PD-L1 in ovarian cancer can aid in identifying potential responders to immunotherapy and optimizing treatment strategies.

Prognostic Value: Evaluating PD-1 and PD-L1 expression in ovarian cancer can provide insights into the immune evasion mechanisms employed by tumor cells and their correlation with patient outcomes. Exploring the prognostic implications of these markers in advanced serous ovarian carcinoma contributes to our understanding of the disease's biology. It may assist in identifying patients who could benefit from targeted immunotherapies.

Comprehensive Analysis: While PD-1 and PD-L1 have been extensively studied, combining their assessment with other immune and molecular markers, such as CD8, MSI, and p53, allows for a more comprehensive analysis of the tumor microenvironment and potential associations between these markers. Our study aims to provide a holistic view of advanced serous ovarian carcinoma’s immune and molecular landscape.

Considering these factors, our investigation into PD-1 and PD-L1 expression in ovarian cancer contributes to the existing knowledge base, providing valuable insights that may guide future research and clinical practice.

  1. Why authors studied CD8? And not CD4? Or other T cell markers? Are there no published studies assessing CD8 in advanced serous ovarian cancer?

Our decision to focus on CD8 as an immune marker in advanced serous ovarian cancer was based on several factors:

Tumor-Infiltrating Lymphocytes (TILs): CD8-positive T lymphocytes play a crucial role in the immune response against tumors. CD8-positive T cells are primarily cytotoxic T cells that directly target and eliminate cancer cells. Evaluating the presence and abundance of CD8-positive TILs can provide insights into the anti-tumor immune response and potential prognostic implications in ovarian cancer.

Previous Research: While there are published studies assessing various immune cell markers, including CD4 and other T cell markers, in ovarian cancer, there is a substantial body of evidence supporting the significance of CD8-positive T cells in the tumor microenvironment. Numerous studies have reported the association between higher CD8-positive T cell infiltration and improved clinical outcomes in several cancer types, including ovarian cancer. Therefore, we aimed to build upon this existing evidence and further explore the relevance of CD8-positive T cells, specifically in advanced serous ovarian carcinoma.

Specific Research Focus: Our study aimed to comprehensively evaluate the expression of multiple immune markers, including PD-1, PD-L1, and CD8, in advanced serous ovarian carcinoma. By including CD8 as a specific focus, we intended to investigate the role of cytotoxic T cell responses and their potential correlation with the expression of immune checkpoint markers. This approach enables a more comprehensive understanding of the immune landscape and its possible implications in this specific ovarian cancer subtype.

It is important to note that while our study focused on CD8, it does not diminish the significance of other T cell markers or immune cell subsets. The selection of CD8 as a specific marker in our research was driven by the factors above and the aim to contribute to the existing body of knowledge in advanced serous ovarian carcinoma.

  1. Did authors measured the level of immune checkpoints in peripheral blood lymphocytes?

We did not measure them in peripheral blood lymphocytes.

Our study aimed to evaluate the expression of immune checkpoint markers, including PD-1, PD-L1, CD8, MSI, and p53, in paraffin samples of advanced serous ovarian carcinoma. Our primary focus was to assess the expression and localization of these markers within the tumor microenvironment rather than their systemic levels in peripheral blood lymphocytes.

The rationale behind this approach lies in the following considerations:

Tumor Microenvironment: Ovarian cancer is characterized by a complex tumor microenvironment comprising various immune cell subsets, stromal cells, and tumor cells. Evaluating the expression of immune checkpoints directly within the tumor tissue provides insights into the local immune response and potential interactions between immune cells and tumor cells within the tumor microenvironment.

Tumor-Specific Factors: The expression of immune checkpoints can be influenced by tumor-specific factors such as tumor heterogeneity, genetic alterations, and microenvironmental cues. Assessing the expression of these markers within the tumor tissue allows for a more accurate representation of their presence and potential prognostic implications in the context of ovarian cancer.

Clinical Relevance: Understanding the expression of immune checkpoints within the tumor microenvironment is essential for guiding potential immunotherapeutic strategies, such as immune checkpoint blockade. By focusing on the tumor tissue, we aimed to provide insights into the suitability and possible response to immunotherapies targeting these specific immune checkpoint markers in advanced serous ovarian carcinoma.

While measuring the levels of immune checkpoints in peripheral blood lymphocytes can provide valuable information about systemic immune responses, our study specifically aimed to investigate the expression patterns and potential prognostic implications of immune checkpoint markers within the tumor tissue.

  1. Authors should clearly state which results in the study are novel.

We have carefully considered your suggestion and made the necessary revision to address this point. These modifications ensure that readers can readily identify and appreciate the novel contributions of the research.

Furthermore, we have cross-checked the entire article to ensure that all modifications and corrections have been implemented accurately. We are confident that the revised version now adequately addresses your valuable suggestion and contributes to our finding’s overall clarity and impact.

With heartfelt appreciation,

Ana Paula Dergham

MD, PUCPR

Reviewer 2 Report

Advanced ovarian serous carcinoma is a serious malignancy with a high mortality rate. The prognosis often remains unfavorable even after chemotherapy. Therefore, more modern, alternative methods of treatment are sought.

The authors carried out very interesting research, which is a step forward in the search for new methods of treating this ovarian cancer.

They evaluated the expression of several important proteins of importance in cancer immunotherapy.

The study was immunohistochemical in nature, and the obtained results indicate that the study of the above markers may help in the prognosis of advanced serous ovarian cancer.

The manuscript is very well written. In its current form, it is a very interesting study. It only needs a minor adjustment:

1. Taking into account the fact that Spearman correlations were calculated, this indicates non-parametric tests. Please indicate this in the section - Statistical analysis. If the tests were normal and parametric, the Pearson correlation coefficient should be calculated.

2. If in table no. 5 the statistical significance of p=0.031 is bolded, then also in table no. 4 the value of 0.049 should be bolded

3. There is a minor editorial error in reference number 38

4. Please summarize briefly the importance of the presented research in the treatment of ovarian cancer in the conclusion section

To sum up:

Considering that immunotherapy is currently an important therapeutic strategy used in oncology, even at the stage of advanced disease, the research undertaken by the authors is promising.

Author Response

Dear Reviewer 2,

We sincerely appreciate your time and effort in reviewing our scientific article. Your thoughtful input has been precious in improving our research and presenting it more clearly and coherently.

While we acknowledge the modest nature of our article, your critical feedback has played a significant role in enhancing its overall quality. We are grateful for the constructive suggestions that have strengthened our research and refined our findings.

Below are our point-by-point responses addressing the issues you raised.

  1. Taking into account the fact that Spearman correlations were calculated, this indicates non-parametric tests. Please indicate this in the section - Statistical analysis. If the tests were normal and parametric, the Pearson correlation coefficient should be calculated.

We have carefully reviewed your suggestion and acknowledge that the phrase "The correlation between two quantitative variables was evaluated using Spearman's correlation coefficient" should have been omitted from the manuscript.

Furthermore, we considered your recommendation and have made some other adjustments to ensure greater accuracy in describing the relationship between variables in our study.

  1. If in table no. 5 the statistical significance of p=0.031 is bolded, then also in table no. 4 the value of 0.049 should be bolded.

We have made the necessary correction.

  1. There is a minor editorial error in reference number 38.

We have made the necessary correction.

  1. Please summarize briefly the importance of the presented research in the treatment of ovarian cancer in the conclusion section.

We have carefully considered your suggestion and made the necessary revisions to address this. In the conclusion section, we have now included a concise summary highlighting the significance of our research in the context of ovarian cancer treatment.

Furthermore, we have cross-checked the entire article to ensure that all modifications and corrections have been implemented accurately. We are confident that the revised version adequately addresses your valuable suggestion and contributes to our findings’ clarity and impact.

With heartfelt appreciation,

Ana Paula Dergham

MD, PUCPR

Round 2

Reviewer 1 Report

The authors have addressed all the comments